# Can We Replicate the Most Demanding Periods of Official Football Matches in Large-Sided Training Games?

**DOI:** 10.3390/jfmk10040410

**Published:** 2025-10-21

**Authors:** David Casamichana, Iñaki Ulloa, Oier Agirrezabalaga, Ibon Etxeazarra, José Manuel González de Suso, Alfonso Azurza, Bixen Calzon, Jon Mikel Arrieta, Iñigo Sasiain, Jon Ollora, Eider Barba, Beñat Erkizia, Aitor Lazkano, Filipe Manuel Clemente, Julen Castellano

**Affiliations:** 1Real Sociedad Institute, Real Sociedad de Fútbol S.A.D., 20160 Donostia-San Sebastián, Spain; david.casamichana@realsociedad.eus (D.C.); inaki.ulloa@realsociedad.eus (I.U.); oier.agirrezabalaga@realsociedad.eus (O.A.); ibon.etxeazarra@realsociedad.eus (I.E.); josemanuel.gonzalezdesuso@realsociedad.eus (J.M.G.d.S.); alfonso.azurza@realsociedad.eus (A.A.); bixen.calzon@realsociedad.eus (B.C.); jonmikel.arrieta@realsociedad.eus (J.M.A.); inigo.sasiain@realsociedad.eus (I.S.); jon.ollora@realsociedad.eus (J.O.); benat.erkizia@realsociedad.eus (B.E.); aitor.lazkano@realsociedad.eus (A.L.); 2Research Group GIKAFIT, Faculty of Education and Sport, University of the Basque Country (UPV/EHU), 01006 Vitoria-Gasteiz, Spain; eider1907@gmail.com; 3Escola Superior Desporto e Lazer, Instituto Politécnico de Viana do Castelo, Rua Escola Industrial e Comercial de Nun’Álvares, 4900-347 Viana do Castelo, Portugal; filipe.clemente5@gmail.com; 4Faculty of Physical Culture, Gdansk University of Physical Education and Sport, 80-336 Gdańsk, Poland; 5Sport Physical Activity and Health Research & Innovation Center, 4900-347 Viana do Castelo, Portugal

**Keywords:** soccer, worst-case scenario, effective playing time, monitoring, GPS, official match

## Abstract

**Objective:** This study aimed to compare the external load demands of large-sided games, with and without regulatory interruptions, to the average (FOOTmatch) and most demanding scenarios (MDSmatch) observed in official football matches, taking into account players’ pitch positions. **Methods:** Large-sided training games were implemented with regulatory interruptions (FOOTtask) and without interruptions (MURDtask), during which a coach continuously introduced new balls to maximise effective playing time. GPS data were collected from eight official matches and six training sessions, involving a total of 30 young male football players. **Results:** MURDtask surpassed FOOTmatch, FOOTtask, and even MDSmatch in distance covered (Effect Size = 0.6–2.5), and equaled MDSmatch in the number of decelerations > 3 m·s^−2^. Both FOOTtask and MURDtask exceeded the average match demands in most locomotor external load variables, except for sprinting efforts > 28 km·h^−1^. However, none of the training games reached the peak values of MDSmatch in high-speed running or high-intensity accelerations and decelerations (>4 m·s^−2^). Positional analysis revealed significantly higher values in MDSmatch for most variables, except for decelerations <−3 m·s^−2^. Conclusions: MURDtask is a useful strategy to overload players, but additional tasks are required to replicate high-speed and high-intensity accelerations and decelerations typical of MDSmatch.

## 1. Introduction

The use of drill-based games, like those related to sided games, is often employed in a training context for augmenting specific behaviors of players while preserving specific dynamics occurring in the games [1]. Among sided games, large-sided formats (e.g., >7v7) (LSGs) are used to prepare soccer players for the physical demands of matches (e.g., high-intensity running and high-speed running), while also developing key tactical principles that promote team unity and effective use of space [2]. Previous studies have shown that this type of practice replicates the average locomotor demands of official matches [3], except in the peak speed, where the values are higher in official matches [3,4]. Previous research [5], which used average metabolic power to measure physical demands, found that large-sided training games even exceed the average requirements of official matches, replicating the stress of the most demanding scenarios (MDSs) of the official match [5]. However, when only the effective playing time is considered, the average demands of the official match are lower than the periods of maximum activity for the players, representing almost 3–4 times the average activity of the official match in variables associated with high-intensity actions [6].

Different studies have been published to compare the activity of players during training tasks in comparison with the MDS of official matches [5,7,8,9]. In general, the results of the studies are consistent in their findings: small-sided games (e.g., 1v1 to 4v4) exceed the values of the MDS of the official match in variables related to acceleration and deceleration. At the same time, medium- and large-sided training games do not reproduce the demands in total distance covered, especially in high-speed running and sprint movements of the MDS of the official match [5,9,10]. More specifically, in terms of analysis of large-sided training games, no research study has reported physical demands that exceed the values of the MDS in variables such as distance covered and distance covered at high speed or sprint. Thus, specifically, Dios-Álvarez et al. [7] found that large-sided training games (300 m^2^ per player) reproduce the average distance covered in official matches but present lower values than the MDS of 1, 5 and 15 min of the official matches analysed. Regarding high-speed running and sprinting movements, the differences are even more pronounced, with significantly higher values observed in the MDS of official matches across all player positions compared to the demands of the tasks studied. The authors therefore concluded that large-sided training tasks do not fully prepare players for the running requirements of official matches [8].

Consistent with the academic literature highlighting positional differences in players’ physical demands during official matches [11], the peak demand scenarios, independent of the window duration, follow a similar pattern [12]. Specifically, central and wide midfielders, as well as full-backs, experience higher peak demands in total distance (TD) covered during football matches compared to forwards and central defenders. In the training process, the differences between player positions are similarly pronounced [10]. Taking distance as the criterion variable, midfielders and wide midfielders covered more total distance and less sprint distance (>7.0 m·s^−1^, m·min^−1^). These positions also exhibited more high-intensity accelerations and decelerations when average metabolic power was used as the criterion variable [10]. Another recent contribution [13] warned that not all types of tasks (small-, medium- and large-sided games) demanded the same total distance and mechanical load, with the smaller number of players per team demanding a more remarkable similarity in physical performance between player positions in absolute terms (distance covered per min and player load per min). However, when relativized to the individual demands of the official match (in percentage), it was the formats with more players and larger dimensions, where similar relative values of physical response between player positions were observed. In this context, it would be valuable to investigate whether large-sided training games can increase physical demands by differentiating player positions in football.

Analyzing peak demands in both large-sided games and match play is crucial for optimizing training prescription in football, particularly when considering playing position. Existing research highlights the positional variations in physical demands during matches [10,14], with specific roles requiring distinct movement patterns and intensities (e.g., midfield roles exhibiting higher high-intensity running demands compared to central defenders). However, current training methodologies often fail to adequately replicate these positional-specific peak demands observed in matches [9]. This gap is particularly evident in the use of LSGs, which, while offering a platform for tactical development and replicating game-like scenarios, may not fully overload players in a manner consistent with their positional peak match demands.

Analyzing the most demanding passages of play for each position in both LSGs and official matches allows us to identify discrepancies in physical demands and develop targeted training interventions. This enables coaches to design LSGs that more accurately simulate the peak demands experienced by players in their respective positions during matches, resulting in more effective and position-specific training programs. Therefore, this study aimed to compare the physical demands of LSGs with and without regulatory interruptions to both the average demands and the most demanding sequences (MDSs) in official matches, taking playing position into account. Our hypothesis was that LSGs without regulatory interruptions can replicate the external load values of the MDS observed in official matches, specifically in terms of effort intensity, exceeding the average external load values of a match while respecting the variations across playing positions.

## 2. Materials and Methods

### 2.1. Participants

Thirty male football players participated in this study (age: 16.7 ± 0.7 years; stature: 177.7 ± 5.7 cm; body mass: 66.8 ± 7.2 kg; sum of the six skinfolds: 34.6 ± 4.9 mm; 30–15 intermittent fitness test: 20.6 ± 1.0 km·h^−1^). The sample excluded players who did not complete all study activities. Players belonged to an under-17-year-old team in a Spanish professional football club. Goalkeepers were excluded from the analyses due to their specific role and training. All of them received a clear explanation of the study, including the potential risks and benefits of participation, and written consent was obtained. Data arose as a condition of the players’ employment whereby they were assessed on a daily basis. Moreover, this study conformed to the Declaration of Helsinki, and once the data were anonymized, the club authorized their use for research purposes. The authorization was required from an institutional ethics committee (code: M10-2024-124).

### 2.2. Design

The large-sided training games (11 vs. 11) in a football field (103 × 63 m, length and width, respectively) were studied both with and without regulatory interruptions. The large-sided training game with regulatory interruptions (FOOTtask) was played with regular interruptions, except for corner kicks, replaced by throw-ins (a common practice for teams during training sessions to give continuity to the game). In addition, the balls were placed outside the field. In the large-sided training game without regulatory interruptions, the Murderball task (MURDtask), one coach was on the field with balls in hand, introducing balls every time there was a regulation interruption, so that an attempt was made to maximize the effective playing time during this large-sided training game.

The number of records in each player position (in FOOTtask and MURDtask, respectively) were: central defender (CD), 48 and 48 files; full-back (FB), 48 and 48 files; central midfielder (CM), 24 and 24 files; attacking midfielder (AM), 42 and 48 files; winger (WN), 48 and 44 files; forward (FW), 26 and 24 files.

During the study period (from October to December), eight official matches were monitored and analysed through GPS devices. For each official match, the player’s average activity in the first half was calculated (FOOTmatch, 80 individual files analysed). The analysis was conducted during the first half, as during this period all players completed the full playing time (no substitutions occurred). Previous studies have shown that the values reached in the MDS during the first halves are higher than those obtained during the second halves [15]. Through the rolling average method applied to each player in the first half of the match, the MDSmatch of the eight external load variables analysed in the study was consistently applied to a time window of 5 min duration (MDSmatch, 80 individual files analysed). This rolling average method entails computing averages for a designated window or interval of consecutive data points as the window progressively moves through the data set. This approach is frequently employed when assessing MDS [16] and has been previously used in several studies [10].

The number of records in each player position was: CD: 16 and 16 files; FB: 16 and 16 files; CM: 8 and 8 files; AM: 42 and 48 files; WN: 16 and 16 files; FW: 8 and 8 files, in FOOTmatch and MDSmatch, respectively.

The external training loads were measured using a 10 Hz GPS that integrated a 100 Hz triaxial accelerometer (WIMU PRO^®^, Realtrack Systems SL, Almeria, Spain). These devices, along with their measurements, demonstrate validity and reliability when employed for GPS-based time–motion analysis in football [17]. It is worth noting that these devices have been bestowed with the FIFA Quality Performance certificate, further attesting to their efficacy and adherence to rigorous standards. Players wore the GPS devices from the beginning to the end of each session. The GPS device was fitted to the upper back (i.e., between the shoulder blades) of each player in a specially designed neoprene harness to minimize movement artefacts. After each session, GPS data were downloaded using the proprietary software package (WIMU SPro V.980, Almeria, Spain) on a personal computer for further analyses. The datasets included in the study achieved a GNSS quality of 71.3 ± 4.4%, an HDOP of 0.76 ± 0.07, and an average of 11.4 ± 0.1 connected satellites. In addition, all GPS recordings were visually examined to identify and remove any potential anomalies or implausible values prior to analysis.

External loads were analysed using the following variables: total distance covered (TD, in m·min^−1^), distances covered at >21 (TD21), >24 (TD24) and >28 (TD28) km·h^−1^ (in m·min^−1^), number of accelerations at >3 (ACC3) and >4 (ACC4) m·s^−2^ (in n·min^−1^), and number of decelerations at <−3 (DEC3) and <−4 (DEC4) m·s^−2^ (in n·min^−1^). All external load variables were analysed relative to each min of practice. The specific thresholds were set in line with previous research studies [18,19]. The inclusion of the high-speed range (TD28) is justified by the increasing attention it is receiving in the context of player preparation [20].

### 2.3. Procedures

The study was conducted in the 2023–2024 competitive season, from October to December. Data collection was carried out during the in-season, in competitive microcycles, keeping environmental conditions such as temperature and humidity similar in all records. The data were collected by an experienced physical coach. The weekly training routines and competitive matches were the usual ones of the competitive training microcycles carried out during the whole season. The studied tasks (large-sided training games) took place in the Wednesday training session, after a standardized general and specific warm-up, approximately 20 min long. Six study sessions were carried out, alternating the large-sided training formats both with and without regulatory interruptions. In sessions 1, 3 and 5, large-sided training games were performed without regulatory interruptions (MURDtask), while in sessions 2, 4 and 6, large-sided training games were performed with regulatory interruptions (FOOTtask). Within each session, four repetitions of 5 min duration of the chosen large-sided training game format were performed, with 6 min of passive rest between repetitions. Water ingestion was allowed during the rest period between repetitions of the large-sided training games chosen. All the large-sided training games were performed outdoors, on an artificial grass pitch and at the same time of day (17:00 h) under similar weather conditions. The motivational instructions provided by the coaching staff were standardized across all training sessions. Coaches were instructed to offer only general verbal encouragement, avoiding any tactical or strategic feedback, to ensure consistency and minimize potential bias in players’ physical responses. During all the repetitions studied, the teams were configured in a 1-4-3-3 game system because it was the structure most used by the team in official matches. The configuration of the teams was carried out by the coaches seeking a balanced performance between the teams, and the teams were kept constant throughout the entire study, except for minor changes when a player was injured.

### 2.4. Statistics

Descriptive statistics were calculated and reported as mean and standard deviation (±SD) for each player position during the large-sided training and official match formats analyzed, for each external load variable. Differences between player positions and activity types in all dependent variables were examined using repeated measures analysis of variance (ANOVA). Post hoc analyses were performed using Bonferroni’s honest significant difference test. Differences between player positions within each large-sided training game format and match format and between activity type for the same player position were assessed via standardized mean differences (Cohen’s d with 90% confidence limits). The interpretation thresholds for standardized effect size (ES) were as follows [21]: <0.2 (trivial), 0.2–0.6 (small), 0.6–1.2 (moderate), 1.2–2.0 (large), and >2.0 (very large). The JASP version 0.18.3.0 (University of Amsterdam, https://jasp-stats.org/, Amsterdam, The Netherlands) was used to conduct the analysis. The statistical significance was set at *p* < 0.05.

## 3. Results

Table 1 describes the average values ±SD of the external load variables for each activity type. In all external load variables, the MDSmatch described higher values than the rest of the activity types in the variables TD21, TD28, ACC3, ACC4 and DEC4 (*p* < 0.05, F = 252.6 and η^2^*p* = 0.68).

Figure 1 describes the distance covered (m·min^−1^) for each player position and type of activity analysed.

Figure 2 shows thirty-six significant differences (*p* < 0.05, F = 68.9 and η^2^*p* = 0.25) were found in the comparison between player positions (intra-task) and intra-player position (inter-task).

Table 2 describes the mean values and standard deviations of the external load variables for each activity type in relation to the player position. In almost all player positions, in the variables of high-intensity sprint (TD28) and neuromuscular (ACC3, ACC4 and DEC4), the MDSmatch task described significantly higher values than the rest of the tasks (*p* < 0.05), with hardly any differences between them.

## 4. Discussion

This study aimed to compare the physical demands of large-sided training games with (FOOTtask) and without (MURDtask) regulatory interruptions with the average demands and the MDS in official matches, considering the position occupied by young football players. The results of this study indicate that MURDtask exceeds even the MDSmatch of the official match in the distance covered (m·min^−1^) and replicates the number of decelerations >3 m·s^−2^. Both large-sided training games, FOOTtask and MURDtask, exceed the average match demands for most external load variables studied; however, none of the training tasks analysed reached the values of the MDSmatch in the variables most associated with high velocity and intensity (high-intensity sprint movements >28 km·h^−1^ and high-intensity accelerations and decelerations > 4 m·s^−2^). All demarcations replicated the same physical pattern for all four tasks: that is, they adjusted their response proportionally to the level of demand of the task. The central hypothesis was partially fulfilled because although the large-sided training game without regulatory interruptions (MURDtask) exceeded the mean values of the official match (FOOTmatch) in all variables, it only outperforms the TD variable in the case of MDS of official matches (MDSmatch). The novelty of this study lies in directly contrasting training formats that manipulate game interruptions (FOOTtask vs. MURDtask) against the most demanding match scenarios, revealing how regulatory pauses influence the players’ ability to replicate or exceed competitive peak loads. This comparison provides new insights into how task design can be optimized to reproduce match intensity and to implement more targeted load periodization strategies in youth football.

It is known that intervening via possession games (without fixed roles) or position games will condition the conditional responses of players [22]. Possession games show greater total distance, peak speed, and player load compared to position games. On the other hand, position games may elicit higher demands of accelerations and decelerations (especially maximal ones). When position games are conducted on larger fields, they tend to yield higher values in distance covered, high-speed running, peak velocity, player load, and maximal acceleration/deceleration [22]. Moreover, when examining the demands relative to player positions, only in position games (such as those in the present study) are differences observed in demands depending on the player’s role, both in official match analyses (e.g., FOOTmatch and MDSmatch) and in the training tasks analyzed (e.g., FOOTtask and MURDtask), as reported in previous studies [8]. In large-sided training games, the intra-task variability—which aligns with the variability seen in official matches—reinforces the principle of training specificity even when considering absolute load values [13,22]. Additionally, by relating these demands to the MDS of official matches, an attempt is made to assess compliance with the overload principle [2,8,23] to determine whether the physical demands of official matches can be effectively replicated in these types of training tasks. 

The demands of large-sided training games have been studied by numerous studies [1,2,23,24,25,26]. In most cases, authors find a greater total and at high-speed running distances covered as the dimensions of the pitch increase [2,8,23,26]. Recent studies suggest that playing areas of at least 250 m^2^ per player are necessary to replicate the total distance covered in MDS of official matches [27]. The surface area per player in the present study respected the dimensions used in official matches (e.g., approximately 325 m^2^ per player), which confirms that the FOOTtask in training replicated the distance covered (m·min^−1^) in the MDS of the official match. However, the MURDtask exceeded the distance covered in the MDS of official matches due to the increased effective playing time resulting from the rule that quickly reintroduces the balls once they go out of play.

High-speed displacement (>21 km·h^−1^) as well as sprinting have proven to be a key element in the preparation of players, very present in goal situations [28] and able to increase the fitness status of the players [29] and protect them from future injury [30], respectively. Regarding player positions, high-speed displacements (>21 and >24 km·h^−1^) in the training tasks (FOOTtask and MURDtask) seem to reproduce the inter-position variability of competitive football, with CD and CM presenting the lowest values [31]. This replicates the results described previously [10]. With respect to running at high-intensity sprints (TD28), it has been suggested that incorporating periodic stimuli into the training process is relevant, allowing players to cover distances at very high speeds and perform sprints [32]. When we analyse high-intensity sprint actions (>28 km·h^−1^), we can observe that the values obtained in the training tasks are very low and without differences between player positions, and well below those achieved by the players in the MDSmatch. Similar results have been obtained by previous studies comparing MDS and different training tasks [8,27]. It seems, therefore, that these training tasks do not reproduce what happens in these periods of maximum demand in terms of this type of movement. We will have to include other tasks in the training or a bigger area per player (>500 m^2^ per player) than those of football (≈300 m^2^ per player) to reproduce, or even supersede, the values achieved by the players in the MDSmatch and attend to positional variability if possible [27].

On the other hand, football players need to constantly perform accelerations (ACC) and decelerations (DEC) during the match [33] and, on some occasions, usually important for the game, at high intensity from low speed or from low intensity to high running speed [34]. The relationship between the frequency of accelerations and decelerations and the dimensions of pitch tends to be negative [19]. That is, the frequency of acceleration and deceleration actions is greater when the dimensions of the pitch are smaller [27]. It is estimated that to reproduce the demands of the players in the MDS of the official matches, the area per player should be between 100 and 150 m^2^ per player [27]. A novel aspect highlighted in the present study is that high-intensity accelerations (ACC3 and ACC4) and decelerations (DEC3), as well as very high-speed running (TD28), are significantly higher in MDSmatch compared to other tasks or activity types. This pattern is consistent both in team averages and across all individual player positions. Therefore, other training proposals, such as small-sided games or tasks that require high-intensity accelerations from low starting speeds [19], should be incorporated into training to stimulate the acceleration/deceleration capacities observed in all player positions during MDSmatch, with minimal differences between positions.

This study provides valuable insights for coaches and practitioners; however, some limitations should be taken into account. Firstly, the sample analyzed consisted of a single team of young football players, which limits the generalizability of the findings to senior or elite populations. Secondly, the reference for the most demanding scenarios (MDS) in official matches was based solely on the analysis of the first halves of eight matches. This approach may underestimate MDS values due to the fatigue effects that typically occur during the second halves of matches. While the study focuses on external load, it is crucial to acknowledge that these measures are a result of the players’ actions and interactions within the game. Analyzing the tactical behaviors and decision-making processes that lead to these external loads would offer a deeper understanding of (e.g., how different game formats influence player effort, why certain positions exhibit higher demands). A promising approach in future research could involve transforming positional data obtained from tracking devices into collective behavioral indicators, such as team surface area, inter-player distances, or spatial dispersion. This would help to link players’ physical responses with tactical behaviors more precisely, providing a more holistic understanding of game dynamics. Additionally, monitoring internal load variables (such as heart rate, lactate levels, and rating of perceived exertion), as well as technical and tactical aspects, would have provided a better contextualization of the tasks proposed in training compared to official match activity, particularly regarding MDS during official matches.

## 5. Conclusions

The main conclusion of this study is that the implementation of specific rules aimed at promoting intra-task continuity in large-sided game formats can elicit enhanced physical responses from players, especially in the locomotor domain. For this reason, it seems advisable that coaches consider implementing this type of task during competitive microcycles to meet the particular needs of each player position and to contextualize the task to the day of the week and period of the season to optimize their performance.

## 6. Practical Applications

The findings of this study offer valuable insights for football coaches aiming to optimize training to better prepare players for the high-intensity demands of competitive matches: (1) se MURDtask to overload locomotor demands and replicate match-like total distance, thereby improving players’ endurance and overall match readiness; (2) complement training with small-sided games or sprint-based drills to specifically target high-speed running and acceleration/deceleration demands that MURDtask alone does not replicate; and (3) integrate MURDtask strategically within the weekly periodization, as it is most effective in the mid-week phase (e.g., acquisition days) when the aim is to stimulate physical load while allowing recovery before competition. Coaches should integrate small-sided games or specific drills emphasizing high-speed running and rapid changes in direction to address this gap.

The study also highlighted positional differences in physical demands, with certain positions (e.g., central midfielders and full-backs) experiencing higher peak demands. Training programs should be tailored to the specific needs of each position to ensure all players are adequately prepared. By implementing these findings, coaches can create a comprehensive and effective training regimen. The use of MURDtask can enhance overall fitness and match readiness, while supplementary drills target high-intensity demands and positional specifics. Integrating large-sided games without interruptions and high-intensity specific drills bridges the gap between training and competitive match demands, leading to better-prepared and more resilient players.

## Figures and Tables

**Figure 1 jfmk-10-00410-f001:**
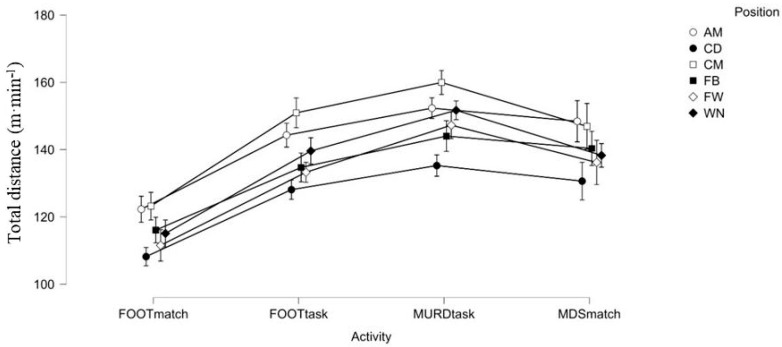
Mean values and standard deviation of total distance (m·min^−1^) in relation to players’ demarcations and activity types. Note: AM is attacking midfielder; CD is central defender; CM is central midfielder; FB is full-back; FW is forward; WN is winger; FOOTtask is the large-sided training game with regulatory interruptions; MURDtask is the large-sided training game without regulatory interruptions; FOOTmatch is official match average; and MDSmatch is match most demanding scenarios.

**Figure 2 jfmk-10-00410-f002:**
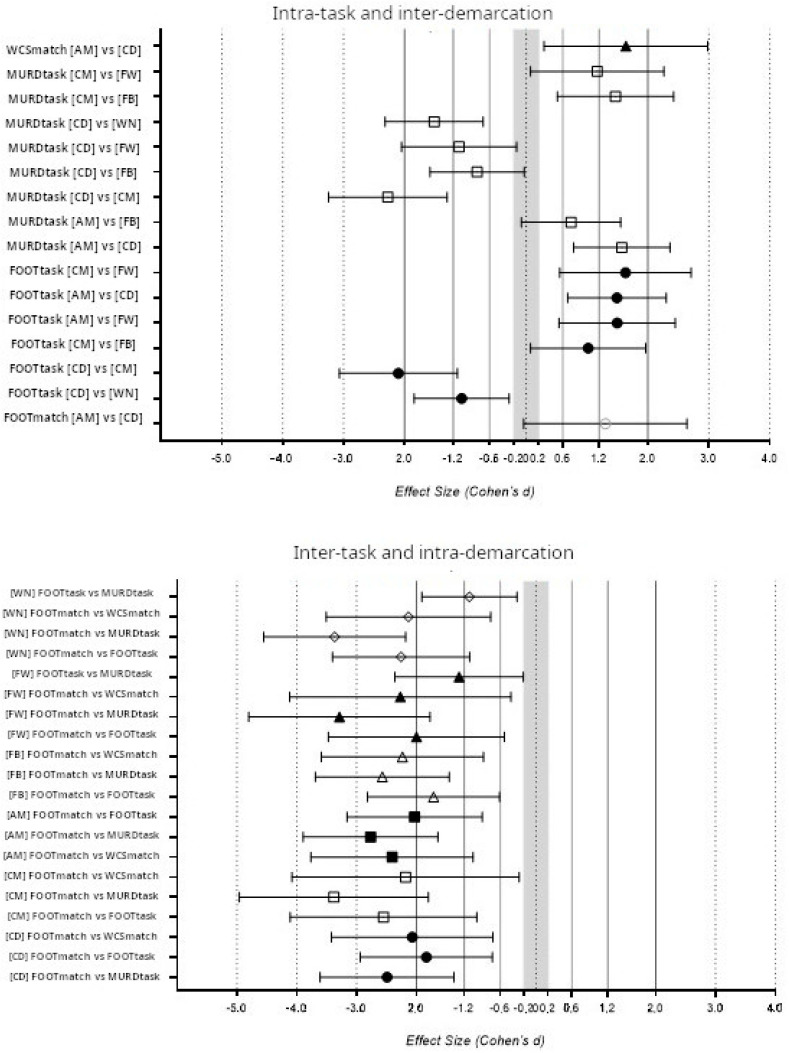
Effect sizes of total distance (TD) in comparison among players’ demarcations and the type of activity. In the upper panel, there is a comparison between intra-tasks and inter-demarcations, and in the bottom panel, there is a comparison between inter-tasks and intra-demarcations. Only significant differences are shown (*p* < 0.05). Note: AM is an attacking midfielder; CD is a central defender; CM is a central midfielder; FB is a full-back; FW is a forward; WN is a winger; FOOTtask is the large-sided training game with regulatory interruptions; MURDtask is the large-sided training game without regulatory interruptions; FOOTmatch is the official match average; and MDSmatch is the match’s most demanding scenario. In the upper panel, WCSmatch is represented by a black triangle (▲), MURDtask is represented by a white square (□), FOOTtask is represented by a black circle (●), and FOOTmatch is represented by a white circle (○). In the lower panel, WN is represented by a white rhombus (◇), FW is a black triangle (▲), FB is a white triangle (△), AM is a black square (■), AM is a white square (□), and CD is a black circle (●).

**Table 1 jfmk-10-00410-t001:** Average values and standard deviation (±SD) of the external load variables for each activity type.

	Activity				
Variables	FOOTmatch (1)	FOOTtask (2)	MURDtask (3)	MDSmatch (4)	Effect Size (95%CI)
TD (m·min^−1^)	115.8 ± 8.2	137.6 ± 13.7 ^1^	147.3 ± 13.5 ^124^	139.9 ± 11.0 ^1^	(1) vs. (2) = −1.71(−2.08/−1.34) (1) vs. (3) = −2.47(−2.86/−2.09) (1) vs. (4) = −1.89(−2.33/−1.45) (2) vs. (3) = −0.77(−1.02/−0.51) (3) vs. (4) = 0.58(0.24/0.93)
TD21 (m·min^−1^)	5.8 ± 2.3	9.9 ± 6.5 ^1^	11.1 ± 7.2 ^1^	16.6 ± 5.6 ^123^	(1) vs. (2) = −0.64(−0.99/−0.30) (1) vs. (3) = −0.84(−1.19/−0.49) (1) vs. (4) = −1.72(−2.16/−1.28) (2) vs. (4) = −1.07(−1.43/−0.72) (3) vs. (4) = −0.88(−1.23/−0.53)
TD24 (m·min^−1^)	2.5 ± 1.6	7.6 ± 8.2 ^1^	8.8 ± 8.9 ^1^	8.9 ± 4.3 ^1^	(1) vs. (2) = −0.68(−1.03/−0.33) (1) vs. (3) = −0.83(−1.18/−0.48) (1) vs. (4) = −0.85(−1.27/−0.43)
TD28 (m·min^−1^)	0.5 ± 0.5	0.6 ± 1.6	0.6 ± 1.3	2.9 ± 2.7 ^123^	(1) vs. (4) = −1.52(−1.95/−1.09) (2) vs. (4) = −1.45(−1.81/−1.09) (3) vs. (4) = −1.46(−1.82/−1.10)
ACC3 (n·min^−1^)	0.7 ± 0.2	0.9 ± 0.4	0.9 ± 0.5	1.7 ± 0.6 ^123^	(1) vs. (4) = −2.20(−2.65/−1.75) (2) vs. (4) = −1.92(−2.29/−1.54) (3) vs. (4) = −1.79(−2.16/−1.42)
DEC3 (n·min^−1^)	0.9 ± 0.2	1.6 ± 0.9 ^1^	2.1 ± 1.2 ^12^	1.9 ± 0.5 ^1^	(1) vs. (2) = −0.74(−1.09/−0.39) (1) vs. (3) = −1.22(−1.57/−0.86) (1) vs. (4) = −1.02(−1.44/−0.59) (2) vs. (3) = −0.48(−0.72/−0.23)
ACC4 (n·min^−1^)	0.1 ± 0.1	0.1 ± 0.2 ^1^	0.1 ± 0.2	0.6 ± 0.3 ^123^	(1) vs. (4) = −2.32(−2.77/−1.86) (2) vs. (4) = −2.35(−2.73/−1.96) (3) vs. (4) = −2.39(−2.77/−2.00) (1) vs. (2) = −0.46(−0.81/−0.12)
DEC4 (n·min^−1^)	0.3 ± 0.1	0.4 ± 0.3	0.4 ± 0.3	0.8 ± 0.3 ^123^	(1) vs. (4) = −2.02(−2.46/−1.57) (2) vs. (4) = −1.55(−1.92/−1.19) (3) vs. (4) = −1.70(−2.06/−1.33)

Note: TD: total distance per min (m·min^−1^); TD21: total distance > 21 km·h^−1^ (m·min^−1^); TD24: total distance > 24 km·h^−1^ (m·min^−1^); TD28: total distance > 28 km·h^−1^ (m·min^−1^); ACC3: number of accelerations > 3 m·s^−2^ (n·min^−1^); DEC3: number of decelerations < −3 m·s^−2^ (n·min^−1^); ACC4: number of accelerations > 4 m·s^−2^ (n·min^−1^); DEC4: number of decelerations < −4 m·s^−2^ (n·min^−1^). 1 is >FOOTmatch; 2 is >FOOTtask; 3 is >MURDtask; 4 is >MDSmatch.

**Table 2 jfmk-10-00410-t002:** Descriptive values (mean ± SD) of the external load variables for each demarcation and type of activity.

	Players	Activity				
Variable	Demarcations	FOOTmatch	FOOTtask	MURDtask	MDSmatch	ES (Range)
TD21	AM	5.2 ± 1.4	8.9 ± 5.4	8.9 ± 6.1	15.9 ± 4.4 ^1^	1.86
	CD	4.1 ± 1.4	7.4 ± 5.9	9.0 ± 6.1	13.1 ± 4.6 ^1^	1.58
	CM	3.5 ± 1.1	6.3 ± 4.3	7.1 ± 5.0	10.4 ± 2.4	
	FB	7.2 ± 1.9	12.9 ± 6.9 ^a,b,d^	14.4 ± 8.4 ^1,b^	19.4 ± 5.0 ^1,2^	1.14–2.12
	FW	8.1 ± 1.8	7.5 ± 4.8	12.0 ± 6.6	21.3 ± 6.3 ^1,2,3,b^	1.61–2.39
	WN	6.9 ± 2.3	13.3 ± 6.4 ^1,a,b,d^	13.9 ± 6.7 ^b^	18.8 ± 4.3 ^1^	1.11–2.07
ES (range)			0.94–1.21	1.19–1.27	1.89	
TD24	AM	1.9 ± 1.3	6.2 ± 7.9	5.7 ± 6.4	7.8 ± 3.8	
	CD	1.6 ± 0.8	5.9 ± 6.9	7.8 ± 7.9	7.4 ± 3.6	
	CM	0.8 ± 0.7	3.4 ± 5.4	4.6 ± 4.6	3.4 ± 1.8	
	FB	3.1 ± 1.2	11.5 ± 10.1 ^1,a,b,d^	13.0 ± 11.4 ^1,e,b^	10.5 ± 2.9	1.17–1.38
	FW	4.1 ± 1.4	4.9 ± 5.0	8.9 ± 8.6	13.2 ± 5.8	
	WN	3.2 ± 1.8	10.4 ± 7.9 ^b^	10.8 ± 9.2	10.7 ± 2.9	
ES (range)			0.78–1.13	1.02–1.16		
TD28	AM	0.3 ± 0.4	0.5 ± 1.2	0.2 ± 0.8	2.1 ± 2.0 ^2,3,b^	1.1–1.28
	CD	0.3 ± 0.3	0.4 ± 1.3	0.4 ± 0.8	1.7 ± 1.4 ^b^	
	CM	0.0 ± 0.1	0.3 ± 0.9	0.0 ± 0.0	0.4 ± 0.7	
	FB	0.6 ± 0.6	1.1 ± 2.6	1.2 ± 2.2	3.6 ± 2.8 ^1,2,3,b^	1.57–2.02
	FW	1.1 ± 0.7	0.3 ± 1.0	0.9 ± 1.5	6.0 ± 3.1 ^1,2,3,b,c^	3.31–3.83
	WN	0.6 ± 0.6	0.6 ± 1.1	0.6 ± 1.1	3.8 ± 2.7 ^1,2,3,a,b^	2.15–2.19
ES (range)					1.42–3.77	
ACC3	AM	0.7 ± 0.1	0.8 ± 0.4	0.7 ± 0.3	1.7 ± 0.7 ^1,2,3^	2.36–2.58
	CD	0.7 ± 0.1	0.7 ± 0.3	0.8 ± 0.5	1.7 ± 0.7 ^1,2,3^	2.13–2.3
	CM	0.6 ± 0.1	0.5 ± 0.3	0.7 ± 0.3	1.4 ± 0.6 ^1,2,3^	1.61−1.98
	FB	0.8 ± 0.2	1.0 ± 0.4 ^b^	1.0 ± 0.4	1.9 ± 0.7 ^1,2,3^	2.12–2.59
	FW	0.8 ± 0.3	1.0 ± 0.4 ^b^	0.9 ± 0.4	1.6 ± 0.7 ^1^	1.87
	WN	0.8 ± 0.1	1.1 ± 0.4 ^a,b,e^	1.3 ± 0.5 ^a,b,e^	1.8 ± 0.6 ^1,2,3^	1.27–2.51
ES (range)			0.77–1.32	1.28–1.49		
DEC3	AM	0.9 ± 0.2	1.7 ± 0.9	2.0 ± 1.1	1.9 ± 0.4	
	CD	0.8 ± 0.2	1.1 ± 0.6	1.7 ± 1.0	1.7 ± 0.4	
	CM	0.9 ± 0.2	1.3 ± 0.6	1.8 ± 1.2	1.8 ± 0.3	
	FB	1.0 ± 0.2	1.8 ± 0.9 ^a^	2.1 ± 1.0	1.9 ± 0.4	
	FW	1.0 ± 0.3	1.8 ± 0.8	2.1 ± 1.0	2.0 ± 1.0	
	WN	1.0 ± 0.3	2.0 ± 1.21 ^1,a^	2.6 ± 1.5 ^1,a^	2.0 ± 0.4	1.09–1.76
ES (range)			0.78	1.01		
ACC4	AM	0.1 ± 0.1	0.1 ± 0.1	0.1 ± 0.2	0.4 ± 0.2 ^1,2,3^	1.89–1.96
	CD	0.1 ± 0.1	0.1 ± 0.2	0.1 ± 0.1	0.6 ± 0.3 ^1,2,3^	2.45–2.59
	CM	0.1 ± 0.0	0.1 ± 0.1	0.1 ± 0.1	0.4 ± 0.2 ^1,2,3^	1.88–2.12
	FB	0.2 ± 0.1	0.2 ± 0.2	0.2 ± 0.2	0.7 ± 0.4 ^1,2,3,e^	2.89–3.06
	FW	0.2 ± 0.1	0.1 ± 0.1	0.1 ± 0.1	0.6 ± 0.2 ^1,2,3^	2.45–2.73
	WN	0.2 ± 0.1	0.2 ± 0.2 ^b^	0.2 ± 0.2	0.6 ± 0.3 ^1,2,3^	2.33–2.49
ES (range)			0.99		1.41	
DEC4	AM	0.3 ± 0.1	0.4 ± 0.3	0.3 ± 0.3	0.8 ± 0.3 ^1,2,3^	1.75–2.15
	CD	0.2 ± 0.1	0.2 ± 0.2	0.3 ± 0.3	0.7 ± 0.2 ^1,2,3^	1.76–1.83
	CM	0.2 ± 0.1	0.4 ± 0.3	0.3 ± 0.3	0.7 ± 0.2 ^1,3^	1.59–1.96
	FB	0.3 ± 0.1	0.5 ± 0.3 ^a^	0.4 ± 0.2	0.9 ± 0.2 ^1,2,3^	1.73–2.31
	FW	0.4 ± 0.1	0.5 ± 0.3 ^a^	0.4 ± 0.3	0.9 ± 0.2 ^1,2,3^	1.88–2.23
	WN	0.4 ± 0.1	0.6 ± 0.3 ^a^	0.5 ± 0.3	0.9 ± 0.2 ^1,2^	1.34–2.21
ES (range)			0.90–1.44			

Note: TD: total distance per min (m·min^−1^); TD21: total distance >21 km·h^−1^ (m·min^−1^); TD24: total distance > 24 km·h^−1^ (m·min^−1^); TD28: total distance > 28 km·h^−1^ (m·min^−1^); ACC3: number of accelerations > 3 m·s^−2^ (n·min^−1^); DEC3: number of decelerations < −3 m·s^−2^ (n·min^−1^); ACC4: number of accelerations > 4 m·s^−2^ (n·min^−1^); DEC4: number of decelerations < −4 m·s^−2^ (n·min^−1^). AM: attacking midfielder; CD: central defender; CM: central midfielder; FB: full-back; FW: forward; and WN: winger. FOOTtask: the large-sided training game with regulatory interruptions; MURDtask: the large-sided training game without regulatory interruptions; FOOTmatch: official match average; MDSmatch: match most demanding scenario. 1 is > FOOTmatch; 2 is >FOOTtask; 3 is >MURDtask; a is >CD; b is >CM; c is >FB; d is >FW; e is >AM.

## Data Availability

The data presented in this study can be accessed upon request from the corresponding author, owing to their confidential and personal nature as properties of the club; access will be granted only if a clearly justified and academically sound motivation is provided.

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
