# Peer review of "Can We Replicate the Most Demanding Periods of Official Football Matches in Large-Sided Training Games?"

_jfmk, 2025, doi:10.3390/jfmk10040410_

Round 1
Reviewer 1 Report
Comments and Suggestions for Authors
Dear Authors
Aim and Background: This article aimed to compare the external load demands of large-sided games, with and without regulatory interruptions, to the average and most demanding scenarios (MDSmatch) observed in official football matches, taking into account players' pitch positions. The Authors conducted a literature review and included this data in their article -different studies. They correctly pointed out studies that compare the activity of players during training tasks with the MDS of official matches.
Method and material:Large-sided training games were implemented with regulatory interruptions. GPS data were collected from eight official matches and six training sessions, involving a total of 30 young male football players.
Results:Statistical calculations were performed taking into account the effect (Effect size = 0.6–2.5). This raises the value of the obtained results.
Formal page of the article: Can we replicate the most demanding periods of football official matches in large-sided training games?
The charts and data in the tables have been correctly presented. This has a very positive impact on the overall text. Both the content of the title and the form (as a question) seem very interesting.
Conclusions: This is a very interesting article in sports science.
Sincerely,
Reviewer
Author Response
We sincerely thank the reviewers for their valuable suggestions and insightful comments, which have significantly contributed to improving the quality and clarity of our manuscript. Their feedback allowed us to strengthen the methodological structure, clarify ethical and data reporting aspects, and enrich both the discussion and the practical applications of the study. We also appreciate the recommendations regarding terminology consistency, justification of analytical thresholds, and the inclusion of updated references, all of which have enhanced the scientific rigor and overall relevance of our work.
Reviewer 2 Report
Comments and Suggestions for Authors
This study on the most demanding periods of football official matches in large-sided training games is relevant to the journal’s scope, and the methodological approach is generally appropriate. However, several issues outlined below require clarification and revision in order to strengthen the manuscript and improve the overall quality of the authors’ work.
1. football official matches should be reordered to official football matches
Regarding introduction
• This introduction is solid in terms of structure and logical flow .
• Avoid long sentences spanning multiple ideas (e.g., lines 76–82). Breaking them up will improve readability. Or correct the punctuation (you may have to sentences as I suggest by the upper letters used - min), However,-
• “Murderball” is introduced at the end (line 103) but without context earlier. Briefly explain earlier in the introduction.
Regarding methods
• Since participants were under 18, further details are needed regarding parental/guardian consent in addition to player consent. Mention if consent form was taken also from the legal guardians.
• You mention six training sessions with alternating formats, but the randomization/order of sessions is not clear. Were sessions counterbalanced or always presented in the same order?
• Please report how data were filtered or cleaned (e.g., were outliers such as unrealistically high accelerations removed?)
• The choice of thresholds is consistent with prior work, but please justify why >28 km·h⁻¹ was used for sprinting (some studies use >25.2 or >30 km·h⁻¹). Reference to consensus guidelines would strengthen this.
• Environmental conditions (lines 172–174) are described as “similar,” but please provide ranges for temperature and humidity.
• Motivational factors: coaches “offered encouragement but not tactical instruction.” Please clarify how standardized this was across sessions to minimize bias
• In Figure 2, “CM is central midfielder” should be corrected to “CM – central midfielder” for consistency, similar to other positions you already mentioned.
• It would strengthen the Results to explicitly state which positional differences were largest
Regarding Results
• In Table 1, the effect sizes are reported with confidence intervals, which is admirable. In Table 2, some ES ranges are reported but others are missing and are presented as ES (range). For transparency and consistency in results interpretation, ensure that only ES or ES + 95% CI are consistently included for all comparisons.
• Sample size per group should be reminded in the table notes because the number of files differs across positions.
• The figure captions currently lack full clarity regarding abbreviations. While Figure 1 includes positional abbreviations (AM, CD, CM, FB, FW, WN), the activity types (FOOTmatch, FOOTtask, MURDtask, MDSmatch) are not explicitly explained in the note. For transparency and readability, please ensure that all abbreviations and terms are fully defined in each figure legend.
Regarding Discussion
• The opening summary is clear, but it could better emphasize the main novel contribution. At present, it reads as a restatement of results (lines 249–262). Strengthen this by explicitly stating what is new about comparing FOOTtask vs. MURDtask relative to MDSmatch, as this is the central innovation.
• Limitations are acknowledged (lines 317–328), but they need more precision: 1) Only first halves of matches were analyzed, which may underestimate MDSdue to fatigue effects in second halves. 2)Sample is single team of young players, generalizability to senior or elite professionals is limited.
• Discussion statements should be more cautious when interpreting overload principle compliance (lines 269–271). While MURDtask exceeded MDS in TD, it did not in sprint or high-intensity accelerations, so overload is not fully achieved.
• The conclusion (lines 330–337) “would enable the overstimulating the players’ physical response” (line 333) is awkwardly phrased.
• The Practical Applications section is valuable but could be more structured and concise. At present, it largely repeats the Results/Discussion in narrative form. I recommend presenting three clear take-home messages for coaches: (1) Use MURDtask to overload locomotor demands and replicate match-like total distance; (2) complement training with small-sided games or sprint drills to specifically target high-speed running and acceleration/deceleration demands; and (3) highlight the periodization aspect more explicitly, noting that MURDtask is most effective in the mid-week (e.g., MD-3 or MD-4). This structure would make the section more impactful and practical for applied settings.
Reference List
• Out of 32 listed references, 17 are post-2018, but more recent studies on GPS-based load monitoring (2022–2025) could be added.
• Please follow the journal’s guidelines. Example: Rico-González, M.; Martín-Moya, R.; Carlos-Vivas, J.; Giles-Girela, F.J.; Ardigò, L.P.; González-Fernández, F.T. Is Cardiopulmonary Fitness Related to Attention, Concentration, and Academic Performance in Different Subjects in Schoolchildren? Funct. Morphol. Kinesiol. 2025, 10, 272.
Author Response

(The authors gave the same response as above.)

Reviewer 3 Report
Comments and Suggestions for Authors
This study examines the potential of large-sided training games (LSG) to simulate the physical demands of official football matches, emphasizing the most demanding scenarios (MDS) and the positional demands on players. The authors compared standard training drills with and without regulatory interruptions—namely, the FOOTtask (with interruptions) and MURDtask (without interruptions)—with match data from eight official games and six training sessions in a sample of U17 male players.
The manuscript offers in offers an important reflection to consider to replicate the matches playing intensity during training sessions. The use of GPS tracking and a rolling average method to identify the most demanding scenario provides a detailed and accurate assessment of peak demands.
The manuscript is well structured and very clear, however I believe it can be further improved with few adjustments in the material and Methods section (from line 108 to line 203), which, being long, could be optimized by introducing at least four subsections: Participants, Design, Procedures, Statistics.
The sentence "Six study sessions were carried out, alternating the large-sided training formats both with and without regulatory interruptions" is repeated both in lines 129-131 and lines 178-179, as well as the meaning of "within each session, four repetitions of 5 min duration of the chosen large-sided training game format were performed .....".
I appreciated the clear explanation of the study's limitations regarding sample and data positional data interpretation
Author Response

(The authors gave the same response as above.)
